# Lung Cancer Immunotherapy: Beyond Common Immune Checkpoints Inhibitors

**DOI:** 10.3390/cancers14246145

**Published:** 2022-12-13

**Authors:** Martina Catalano, Sonia Shabani, Jacopo Venturini, Carlotta Ottanelli, Luca Voltolini, Giandomenico Roviello

**Affiliations:** 1School of Human Health Sciences, University of Florence, 50134 Florence, Italy; 2Thoraco-Pulmonary Surgery Unit, Careggi University Hospital, 50134 Florence, Italy; 3Department of Experimental and Clinical Medicine, University of Florence, 50134 Florence, Italy

**Keywords:** lung cancer, immunotherapy, emerging immune checkpoint inhibitors, novel immune targets

## Abstract

**Simple Summary:**

The lung cancer treatment paradigm has been completely changed by immunotherapy; however, less than half of the treated patients obtain a response, and an even smaller proportion achieve a long survival. Primary and acquired resistance mechanisms and the immune-related adverse events limit the use of available immune checkpoint inhibitors (ICIs), anti-cytotoxic T-lymphocyte antigen 4 (CTLA-4) and programmed death protein 1/ligand 1 (PD-1/PD-L1). Several predictive biomarkers of ICI response have been evaluated so far, but only PD-L1 expression was approved for clinical use. In the last few years, new immune targets have been identified, and both inhibitory and stimulatory treatments have been developed. These molecules have shown to be safe and effective mostly in combination with anti CTLA-4 and PD-1/PD-L1. Preliminary data indicate their activity in non-small- and small-cell lung cancer, thus allowing the scheduling of further studies to improve the still poor prognosis of these patients.

**Abstract:**

Immunotherapy is an ever-expanding field in lung cancer treatment research. Over the past two decades, there has been significant progress in identifying immunotherapy targets and creating specific therapeutic agents, leading to a major paradigm shift in lung cancer treatment. However, despite the great success achieved with programmed death protein 1/ligand 1 (PD-1/PD-L1) monoclonal antibodies and with anti-PD-1/PD-L1 plus anti-cytotoxic T-lymphocyte antigen 4 (CTLA-4), only a minority of lung cancer patients respond to treatment, and of these many subsequently experience disease progression. In addition, immune-related adverse events sometimes can be life-threatening, especially when anti-CTLA-4 and anti-PD-1 are used in combination. All of this prompted researchers to identify novel immune checkpoints targets to overcome these limitations. Lymphocyte activation gene-3 (LAG-3), T cell immunoglobulin (Ig) and Immunoreceptor Tyrosine-Based Inhibitory Motif (ITIM) domain (TIGIT), T cell immunoglobulin and mucin-domain containing-3 (TIM-3) are promising molecules now under investigation. This review aims to outline the current role of immunotherapy in lung cancer and to examine efficacy and future applications of the new immune regulating molecules.

## 1. Introduction

The last decade has seen the rapid development of immunotherapy and its role as a crucial strategy in cancer treatment, particularly in the field of lung cancer. The immune system closely interacts with tumors along the entire process of cancer onset and progression. Tumor cells develop numerous ways to escape immune cell recognition and removal by regulating their antigen presentation, through the secretion of immunosuppressive cytokines such as interleukin (IL)-10 and transforming growth factor (TGF)-β, or by affecting their metabolism, causing an alteration in the tumor microenvironment (TME) [1,2,3,4,5,6]. However, the most potent mechanism to limit normal anti-tumor immune responses is the activation of immune checkpoint pathways such as cytotoxic T-lymphocyte antigen 4 (CTLA-4), programmed death 1 (PD-1) and programmed death ligand-1 (PD-L1) [7,8,9]. In fact, the CTLA-4 and PD-1/PD-L1 blockade was capable of restoring the host’s T cell-mediated immune system response, suppressed by the tumour [9]. These findings paved the way to the development of immune checkpoint inhibitors (ICIs), which take advantage of the host’s immune system to enhance anti-tumor activity. Clinical efficacy and durable responses were recorded in several tumour types [10,11,12,13], especially in non-small-cell lung cancer (NSCLC) [14]. In patients with NSCLC without a driver mutation and in those with small-cell lung cancer (SCLC), immunotherapy in the form of ICIs is currently the cornerstone of treatment [15,16]. In NSCLC PD-L1, despite representing to date the most reliable predictive biomarker of response to immunotherapy, it fails to select the right subset of patients who would benefit from this treatment. Indeed, only a limited number of patients respond to ICI, and also in the event of a lasting response they eventually experience disease progression. Moreover, due to the paucity of effective second-line treatments, the mortality rate of this disease remains still high [17,18,19]. In addition, about 15–25% of patients treated with ICIs developed serious immune-related adverse events (irAEs), which can sometimes be fatal [20,21,22]. Therapy strategies which involve the combination of ICI with each other or with other drugs (i.e., chemotherapy, target therapy, agents, poly ADP ribose polymerase (PARP) inhibitors) or local treatment, have been adopted to overcome these hindrances, resulting in increased clinical responses. However, as many patients show primary or acquired resistance to ICIs [23,24,25,26], a great interest is addressed to discover novel targets. The next generation immune checkpoints, such as lymphocyte activation gene-3 (LAG-3), T cell immunoglobulin (Ig) and Immunoreceptor Tyrosine-Based Inhibitory Motif (ITIM) domain (TIGIT), T cell immunoglobulin and mucin-domain containing-3 (TIM-3), V-domain Ig suppressor of T cell activation (VISTA), B7 homolog 3 protein (B7-H3), inducible T cell costimulatory (ICOS), and B and T cell lymphocyte attenuator (BTLA), appear to be promising therapeutic strategies with the possibility of future clinical applications [27,28,29,30,31,32,33]. Furthermore, the addition of novel ICIs, which do not exhibit overlapping mechanisms of action with those already in use, could improve efficacy and decrease toxicity. Therefore, new efforts are required to strengthen the immune system, expand treatment choices, and delay ICI resistance. In this review, we will summarize the current role of immunotherapy in lung cancer and discuss the potential and future perspectives of new immune targeting targets.

## 2. Current Role of Immunotherapy in Lung Cancer

In the past seven years, numerous ICIs received approval for lung cancer treatment in different settings of disease, particularly for NSCLC. Indeed, ICIs commenced as a second-line treatment strategy for metastatic NSCLC. The therapeutic indications were then extended to the first-line advanced settings and later also to the earlier stages, including both unresectable and resectable disease (Figure 1).

The most recent Food and Drug Administration (FDA) approval of ICI in NSCLC stems from the results of the CheckMate 816 trial [34]. In this study, three cycles of neoadjuvant chemotherapy plus nivolumab (anti-PD-1) in patients with resectable disease resulted in a significantly higher percentage of pathological complete response (pCR) and longer event-free survival (EFS) than chemotherapy alone [34]. The phase III trial IMpower 010 compared atezolizumab (anti-PD-L1) versus best supportive care (BSC) in patients with resectable stable IB-IIIA NSCLC undergoing complete surgical resection and subsequent adjuvant platinum-based chemotherapy [35]. The superiority of atezolizumab in disease-free survival (DFS), allowed its recent FDA approval in adjuvant setting for patients with II-III stage NSCLC, harboring PD-L1 positivity [36].

For unresectable stage III NSCLC, a placebo-controlled phase III trial revealed that treatment with durvalumab for 12 months significantly improves progression-free survival (PFS) (17.2 vs. 5,6 months) and OS (not reached [NR] vs. 28.7 months) for patient with PD-L1 expression and who had not progressed to concurrent chemoradiation. This results were confirmed at 5-year follow up [37,38].

In the advanced NSCLC setting, nivolumab was the first drug to obtain the FDA accelerated approval in 2015 as a second-line treatment after progression to platinum-based chemotherapy. Two-phase III clinical trials, CheckMate 017 and 057, showed the superiority of nivolumab compared with docetaxel in terms of objective response rate (ORR) and overall survival (OS) [39,40]. In the wake of these results, two more ICIs, pembrolizumab (anti-PD-1) and atezolizumab, displayed a comparable efficacy and have been subsequently approved for the second-line setting [41,42].

However, the big breakthrough came with the following phase III clinical trials, which widened the first-line treatment opportunities for patients with metastatic NSCLC. Indeed, ICIs proved to be superior alone or in combination with platinum-based chemotherapy over standard treatment. ICIs not only improved response rates (RR) and recorded the longest OS ever achieved, but ensured also a long-lasting survival benefit. In the KEYNOTE-024 study, pembrolizumab significantly improved PFS and OS in patients with advanced NSCLC and PD-L1 expression on at least 50% of tumor cells [43]. Likewise, atezolizumab and cemiplimab (anti-PD-1), tested in NSCLC patients with high PD-L1 expression, resulted in significantly longer OS than platinum-based chemotherapy [44,45]. Based on these results FDA approved pembrolizumab, atezolizumab, and most recently cemiplimab, for the first line treatment of adult patients with metastatic NSCLC and PD-L1 expression ≥50%, without genomic tumor aberration on epidermal growth factor receptor (EGFR) or anaplastic lymphoma kinase (ALK). 

Many chemo-immunotherapy combinations have been explored, showing to be both efficacious and well-tolerated. The combination of pembrolizumab with carboplatin and pemetrexed received accelerated FDA approval in 2017 based on the phase II study KEYNOTE-021 cohort G [46]. The subsequent phase III trial KEYNOTE-189 confirmed the latter results, by testing pembrolizumab versus placebo plus four cycles of platinum-based chemotherapy and pemetrexed (continued as maintenance therapy), on patients with untreated, nonsquamous, EGFR and ALK wild-type NSCLC [47]. In the experimental arm, the median PFS and OS were 9.0 and 22.0 months versus 4.9 and 10.7 months recorded in the control arm. Pembrolizumab, added to chemotherapy, showed a good safety profile and also improved RR (48.0% vs. 19.4%) and the median duration of response (DOR) (12.4 vs. 7.1 months) [47]. Likewise, pembrolizumab added to platinum-based chemotherapy plus paclitaxel or nab-paclitaxel resulted in significantly longer OS (15.9 vs. 11.3 months) and PFS (6.4 vs. 4.8 months) than chemotherapy alone in untreated metastatic, squamous NSCLC regardless PD-L1 expression, thus receiving FDA approval [48]. Moreover, also atezolizumab obtained FDA approval with the phase III study IMpower 130 for previously untreated metastatic, non-squamous, NSCLC patients [49]. Better median PFS and OS (18.6 vs. 13.9 months) and RR (49.2% vs. 31.9%) were observe in the combination arm compared with the chemotherapy alone group [36]. An additional combination based on cemiplimab plus platinum-doublet chemotherapy has been evaluated in the phase III EMPOWER-Lung 3 study, as first-line treatment for advanced NSCLC, irrespective of PD-L1 expression or histology [50]. Cemiplimab plus chemotherapy recorded a median OS of 21.9 months compared to 13.0 months with chemotherapy alone and has been accepted for review by the FDA.

A further immunotherapy strategy that showed advantage in treating lung cancer is combining PD-1/PD-L1 and CTLA-4 inhibitors. In the phase III CheckMate 227 trial, previously untreated NSCLC patients were randomized to receive in a 1:1:1 radio nivolumab plus ipilimumab, nivolumab alone, or chemotherapy [51]. In PD-L1 positive patients nivolumab plus ipilimumab showed a median OS of 17.1 months compared to 14.9 months with chemotherapy. ICIs combination recorded a higher RR (35.9% vs. 30.0%) and DOR (23.2 vs. 6.2 months) than chemotherapy alone. Despite exhibiting better outcomes than chemotherapy also in PD-L1 negative NSCLC, FDA approved the immunotherapy combination as first-line therapy only for PD-L1 positive disease [52].

Eventually, dual-checkpoint blockade combined with chemotherapy was investigated in a phase III study, CheckMate 9LA, including treatment-naïve NSCLC patients regardless of PD-L1 expression and tumor histology [53]. Patients were randomized to receive chemotherapy alone for four cycles and eventually maintenance with pemetrexed or a combination of nivolumab with ipilimumab every 6 weeks and concurrent platinum-doublet chemotherapy for only two cycles. This new chemo-immunotherapy approach met its primary endpoint, by recording a median OS of 15.6 months in the experimental arm and 10.9 months in the control group. The combination also improved PFS (6.7 vs. 5.0 months) and RR (38% vs. 25%) in the face of a higher grade toxicity (≥3 in 47% vs. 38%) and greater discontinuation rate (19% vs. 7%) compared with chemotherapy alone. The chemo-immunotherapy regimen was recently FDA approved in 2020 [54]. Very recently, FDA approved tremelimumab in combination with durvalumab and platinum-based chemotherapy for patients with metastatic NSCLC with no sensitizing EGFR, ALK genomic tumor aberrations. The phase III POSEIDON study, enrolled patients to receive the following: tremelimumab plus durvalumab and platinum-based chemotherapy for four cycles, followed by durvalumab until progression and one additional tremelimumab dose; durvalumab plus chemotherapy for four cycles, followed by durvalumab; or chemotherapy for up to six cycles ± maintenance pemetrexed [55]. Durvalumab plus tremelimumab and chemotherapy significantly improved PFS (6.2 v 4.8 months) and OS (14.0 v 11.7 months) compared to chemotherapy alone.

Unlike the success achieved in NSCLC, the application of ICIs in SCLC remains limited and not very advantageous. The only innovation relates to the implementation of first line extended-stage SCLC (ES-SCLC) therapy, which saw the addition of two anti-PD1 inhibitors to standard platinum-based chemotherapy. Atezolizumab has been approved in 2019 after phase III trial IMpower 133, which accrued patients with previously untreated ES-SCLC to receive four cycles of standard carboplatin plus etoposide and concurrent atezolizumab or placebo followed by maintenance atezolizumab or placebo [56]. The addition of Atezolizumab to chemotherapy improved median OS (12.3 vs. 10.3 months), PFS (5.2 vs. 4.3 months) and patient’s quality of life, without increase the toxicity. After about a 1 year, durvalumab received a similar approval based on the phase III CASPIAN trial. Patients were randomized to receive standard chemotherapy (platinum plus etoposide) alone or in combination with durvalumab (continued as maintenance therapy) ± tremelimumab. A median OS of 13.0 months versus 10.3 months has been recorded in the experimental arm compare to chemotherapy alone group, with 34% versus 25% of patients alive at 18 months, respectively [44]. These results made the chemo-immunotherapy combination the first intervention capable of improving survival in ES-SCLC over three decades.

### PD-L1 Expression Levels and Outcome Related

PD-L1 is currently one of the few recognized and approved biomarkers predictive of response to immunotherapy. Despite the confirmed benefit of assigning ICIs according to PD-L1 expression, the latter biomarker alone is still inadequate to select the right candidates for immunotherapy [57]. In NSCLC, patients with higher levels of PDL1 expression tend to respond more favorably to the ICIs [58,59]. Different diagnostic immunohistochemistry test, with variations in cut-off values have been used to establish PD-L1 expression [60,61,62,63].

The frequency of PD-L1 expression in lung cancer has been reported by several authors [64,65,66]. In the largest real-world study conducted on 2368 advanced NSCLC patients, 22% had PD-L1 TPS ≥ 50%, 52% PD-L1 TPS ≥ 1%, and 48% PD-L1 TPS < 1%. Prevalence of PD-L1 TPS ≥ 50% and TPS ≥ 1% were similar across geographic regions ranging from 21–24% and 47–55%, respectively [64]. Another study assessed PD-L1 expression in 264 cases of NSCLC showing: high PD-L1 expression (≥50%) in 29.5% of cases, low (1–49%) in 43.9% and absent (<1%) in the 26.5% [66]. Skov et al., in their prospective study, included 819 patients with NSCLC reported a PD-L1 ≥ 1% positive cells in the 63% of NSCLC patients and PD-L1 ≥ 50% in 30% [66].

Unlike NSCLC, in other types of lung cancer such as SCLC PD-L1 expression levels are understudied, with contradictory reports of expression status [67].

Most recently Xu et al. conducted a meta-analysis to evaluate the efficacy of ICI monotherapy or combined with chemotherapy and estimate the predictive value of PD-L1 expression in predicting the response from these treatment [67]. Results showed better OS, PFS and ORR with anti-PD-1/PD-L1 monotherapy compared with chemotherapy in the intention-to-treat population (ITT) and emphasized the value of positive PD-L1 expression in predicting improvement of clinical outcome from anti-PD-1/PD-L1 treatment. Indeed, better efficacy outcomes correlated with higher PD-L1 levels (mainly PD-L1 ≥ 50%), whereas no statistical survival benefit was observed for the PD-L1 < 1% population who received anti-PD-1/PD-L1 monotherapy compared to chemotherapy alone. Subgroup analyzes showed significant improvement in ORR from ICI in patients with PD-L1 ≥ 50%, no difference in patients with PD-L1 < 1%, and better ORR with chemotherapy versus ICI monotherapy in patients with PD-L1 expression ranging from 1 to 49%.

Similar results derived from Liu et al.’s metanalysis [67]. In this study, in patients with PD-L1 ≥ 1%, ten immunotherapy combinations were associated with significantly prolonged OS and PFS (the latter especially with anti-PD-1 plus chemotherapy) compared with chemotherapy. In patients with PD-L1 1–49%, seven immunotherapy combinations also significantly improved OS and PFS compared with chemotherapy. In patients with PD-L1 ≥ 50%, nine immunotherapy combinations (except for durvalumab-tremelimumab), showed significantly higher OS and PFS benefit than standard chemotherapy [67].

Finally, another metanalysis investigated the efficacy and safety of dual ICIs ± other therapies. An improved OS with the combination therapy in the ITT population was shown. However, according to the analysis, no statistically significant difference between the two groups was found for patients with PD-L1 < 1%, thus narrowing the benefit from this combination for PD-L1 ≥ 1% expression [68].

## 3. Newly Immune Checkpoints

In addition to CTLA-4 and PD-1/PD-L1, novel immune checkpoint molecules expressed on T cells have been revealed and are currently under investigation [69]. In this section, we will deepen the mechanism of action of some novel molecules (Figure 2) and their role in regulating immune responses. Moreover, we will present the results currently available from both lung cancer and the ongoing clinical trials.

### 3.1. Lymphocyte-Activation Gene 3

The LAG-3 is a protein composed of four parts: an hydrophobic leader, an extracellular region (consisting in four Ig superfamily-like domains [D1–D4]), and a transmembrane and cytoplasmic region [70]. The D4 transmembrane domain connecting peptide is prone to cleavage by metallo-proteases generating the soluble LAG-3, essential for normal activation of T cells [71,72,73,74]. The major class II histocompatibility complex (MHC-II) is the canonical ligand of LAG-3, which induces the exhaustion of immune cells and the decrease in cytokine secretion [75,76].

There is some evidence to indicate that LAG-3 downregulates the T helper 1 (Th1) cell activation, proliferation, and cytokines secretion [77,78,79]. High levels of LAG-3 expression have been associated with tumor progression, poor prognosis, and unfavorable clinical outcomes in various types of cancer [80]. Similarly, to PD-1, LAG-3 seems to contribute to the immune escape mechanisms in tumors and therefore, has been proposed as a promising therapeutic target. LAG-3 is often simultaneously co-expressed with other ICs, such as PD-1, TIGIT and TIM-3 [81,82]. Studies on murine models showed that dual blockade of LAG-3 and PD-1 improved anti-tumor immune response by increasing CD8+ tumor-infiltrating cells in the TME and decreasing Treg cells [73]. These findings led to clinical trials based on targeting LAG-3 alone or in combination with other ICIs.

A soluble LAG-3 fusion protein, eftilagimod alpha, has been tested in combination with pembrolizumab in patients with NSCLC. Eftilagimod combined with anti-PD-1 was safe and showed encouraging antitumor activity in all comer PD-L1 positive first line NSCLC [83].

Relatlimab is an IgG4 monoclonal antibody (mAb) targeting LAG-3, firstly investigated in patients with unresectable or metastatic melanoma showing significant improvement in PFS when combined with nivolumab compared to nivolumab alone [84]. Currently, relatlimab combined with nivolumab and chemotherapy is under evaluation as first-line treatment of advanced NSCLC in a phase II trial (NCT04623775).

Ieramilimab, an IgG4 mAb anti-LAG-3, has been evaluated in combination with spartalizumab (mAb anti-PD-1) in a phase II study enrolled patients with different tumor types including SCLC relapsed or refractory to standard therapies. The dual blockade of LAG-3 and PD-1 showed promising activity in SCLC with a clinical benefit rate at 24 weeks (primary endpoint) of 0.27 [85]. In another phase, I/II ieramilimab demonstrated that it was well tolerated as a monotherapy and in combination with spartalizumab (mAb anti-PD-1) in patients with advanced solid tumors [86]. Modest antitumor activity was seen with combination treatment and the toxicity profile was comparable to that of spartalizumab alone [86].

REGN3767 alone or in combination with cemiplimab was evaluated in a dose escalation study, in patients with advanced malignancies, showing a tolerable safety profile [87]. Data from four trials investigating BI 754111 (anti-LAG-3 mAb), in combination with BI 754091 (mAb anti-PD-1), in patients with advanced solid tumors showed a manageable safety profile, similar to other ICIs [88]. Similarly, Sym022 binding LAG-3 and blocking the interaction with MHCII, demonstrated tolerable safety profiles alone or in combination with anti-PD-1 or anti-TIM-3 mAb, in patients with solid tumors [89].

MGD013 is a dual-affinity re-targeting protein, designed to bind LAG-3 and PD-1. Immature results showed encouraging early evidence of anti-tumor activity and a satisfactory safety profile with 70.5% of treatment related AEs (most commonly fatigue and nausea), of which 23.2% of grade ≥3 consistent with events observed with anti-PD-1 Abs [90]. The adverse reactions reported with anti-LAG 3 were similar to the ICI ones, with musculoskeletal pain, fatigue, rash, pruritus, and diarrhea described in ≥20% of patients. The combination therapy showed a safety profile analogous to ICI monotherapy [91].

### 3.2. T Cell Immunoreceptor with Ig and ITIM Domains

TIGIT is a protein composed of an extracellular Ig variable domain, a transmembrane domain and a short intracellular domain endowed with one immunoreceptor tyrosine-based inhibitory motif (ITIM) and one immunoglobulin tyrosine tail (ITT)-like motif [29,30,31,32,33,34,35,36,37,38,39,40,41,42,43,44,45,46,47,48,49,50,51,52,53,54,55,56,57,58,59,60,61,62,63,64,65,66,67,68,69,70,71,72,73,74,75,76,77,78,79,80,81,82,83,84,85,86,87,88,89,90,91,92,93,94].

TIGIT belongs to the family of Poliovirus Receptor (PVR)-like proteins and shares sequence homology with other members of the PVR-like family [32,33,34,35,36,37,38,39,40,41,42,43,44,45,46,47,48,49,50,51,52,53,54,55,56,57,58,59,60,61,62,63,64,65,66,67,68,69,70,71,72,73,74,75,76,77,78,79,80,81,82,83,84,85,86,87,88,89,90,91,92,93,94,95]. Similarly to the CTLA-4/B7/CD28 pathway, TIGIT competes with CD266 or CD96 to exert its immune role [96,97]; as CD96 deliver inhibitory signals on T cells, conversely CD226 delivers a positive co-stimulatory signal. Initially, TIGIT was able to suppress T cell activation indirectly, through the bond to CD155 on DCs and the release of IL-10. Nowadays it is supposed that through the competition with CD226, TIGIT can directly suppress T cell functions [93,98]. Regarding the interaction with NK cells, it has been demonstrated that through the cytoplasmic ITIM domain, TIGIT could negatively modulate NK cells both in humans and mice [99,100,101]. Moreover, through the link with Fibroblast activation protein 2 (Fap2), a *Fusobacterium nucleatum* derived protein, TIGIT triggers a negative signal that suppress T and NK cells activities, thus mediating a tumor-immune evasion mechanism [102].

Therefore, TIGIT has been considered an important immune checkpoint able to inhibit several steps of the cancer immunity process, and some trials noted the good therapeutic potential of targeting TIGIT in different tumor types [103]. Evidence suggests that the TIGIT blocking may restore T cell activity in cancer patients and that dual blockade of TIGIT and PD-1/PD-L1 improving synergistically the CD8+ T cells antitumor function in mice, results in an increased protective activity of memory T cells, in a complete tumor rejection, and a prolonged OS [104]. A recently published meta-analysis demonstrated the prognostic value of tumor-infiltrating TIGIT + CD8+ T-cells in patients with solid cancers in which its high expression is associated with a worst OS and relapse free survival (RFS) [105], albeit Fang et al. reported a favorable outcome with longer OS and RFS due to the correlation between CTLA-4 and TIGIT in breast cancer patients [106]. Therefore, TIGIT blockade is a promising immune target and the dual blockade with the anti PD-1/PD-L1 may potentially help overcome the immune-resistance observed with the use of a single ICI.

Tiragolumab, an anti-TIGIT antibody in association with atezolizumab has been evaluated in the CITYSCAPE study, a phase II trial conducted on patients with PD-L1 positive NSCLC [107]. The analysis showed a satisfying safety profile and an improved ORR (31.3 vs. 16.2%) and PFS (5.4 vs. 3.6 months) in the combination treatment with durable responses particularly in patients with a PD-L1 expression score >50%. Due to these results tiragolumab received FDA breakthrough therapy designation, however a confirmatory phase III study, (SKYSCRAPER-01) is currently ongoing in in these patients (NCT04294810). The phase II/III trial SKYSCRAPER-06, comparing atezolizumab plus pemetrexed and platinum-based chemotherapy with or without tiragolumab in patients with previously untreated advanced non-squamous NSCLC, is ongoing (NCT04619797). One phase III trial evaluated atezolizumab and tiragolumab versus durvalumab in patients with locally advanced, unresectable stage III NSCLC (NCT04513925), whereas an ongoing phase II study aimed to compare the effects of neoadjuvant and adjuvant tiragolumab plus atezolizumab, with chemotherapy versus chemotherapy alone, in patients with previously untreated locally advanced resectable stage II, IIIA, or select IIIB NSCLC (NCT04832854). Although in the phase III trial SKYSCRAPER-02, the addition of tiragolumab to atezolizumab and standard chemotherapy does not provide any benefit over atezolizumab and chemotherapy alone in patients with untreated ES-SCLC with or without brain metastasis, the combination was well tolerated, and the final OS analysis will be presented [108]. A phase II study is evaluating atezolizumab ± tiragolumab as consolidation therapy in participants with limited stage SCLC who have not progressed to chemoradiotherapy (NCT04308785).

Another anti-TIGIT mAb, vibostolimab, was studied in a phase I trial in PD-1/PD-L1-naïve patients with refractory advanced NSCLC. The drug exhibited an acceptable toxicity profile and antitumor activity both alone or in combination with pembrolizumab [109]. An ongoing phase III trial is evaluating pembrolizumab alone and with vibostolimab in PD-L1 positive NSCLC patients (NCT04738487).

Domvanalimab is a humanized IgG1 mAb targeting TIGIT, whose combination with zimberelimab (anti-PD-1) is under investigation in a phase III trial on PD-L1-positive locally advanced or metastatic NSCLC patients. The remaining two phase II studies are testing zimberelimab plus etrumadenant (adenosine receptor antagonist) in untreated and treated NSCLC (NCT 04262856, NCT 04791839).

Anti-TIGIT antibodies were overall well tolerated when administered as monotherapy as well as in combination with PD-1/PD-L1 blockers. Most common AEs reported in ≥10% patients included grade 1 fatigue and pruritus, whereas grade 2 anemia and diarrhea were reported in two patients treated with vibostolimab monotherapy. There were no grade ≥3 events reported with anti-TIGIT antibody monotherapy [91].

### 3.3. T Cell Immunoglobulin and Mucin-Domain Containing-3 (TIM-3)

TIM-3 is a protein made up of an extracellular Ig variable region-like domain, a transmembrane domain, and an intracellular cytoplasmic tail with 5 potential tyrosine phosphorylation sites [28]. The phosphorylation mediated by Src kinases or interleukin-inducible T-cell kinase (ITK) on Y265 and Y263 sites, allows the release of HLA-B-associated transcript 3 that is crucial for downstream signaling [110,111,112]. Ig variable region domain of TIM-3 is the target for two soluble ligands, galectin-9 and high-mobility group protein B1 (HMGB1), and two surface ligands, Carcinoembryonic Antigen Cell Adhesion Molecule 1 (CEACAM1) and phosphatidylserine (PtdSer). [113,114]. All these molecules play a role in immunesuppressive pathways. Galectin 9 is capable of inducing Th1 cells apoptosis through an intracellular calcium influx, whereas CEACAM1 acts as a negative regulator of T cell responses. In tumors, TIM-3 competes with nucleic acids in binding HMGB1, highly expressed in tumor-infiltrating DCs, and in reducing their transport to the endosomes, thereby mitigating the innate immune response to tumor-associated nucleic acid [115]. Finally, the interaction of PtdSer with TIM-3, although weaker than the other ligands, plays a role in elimination of apoptotic bodies, helping the antigen cross-presentation [116].

Several pieces of data confirmed the role of TIM-3 in tumor biology and its ability to promote tumor cell proliferation, migration, and invasion [117]. Evidence suggested a negative prognostic role of TIM-3 expression in several types of cancer but its role in clinical cancer trials is still controversial [118]. Recently, a meta-analysis confirmed the negative prognostic value of TIM-3 expression among several tumors including lung cancer, although in the same analysis a favorable role of the TIM-3 expression was observed in other malignancies such us breast cancer and malignant pleural mesothelioma. These data seem to suggest a double prognostic value of TIM-3, depending on the different types of cancers. The ongoing clinical trials are assessing the value of TIM-3 as an immunotherapy target.

Drug combinations targeting both TIM-3 and PD-L1 immune checkpoint pathways were evaluated in a phase Ia/Ib dose escalation study; patients with advanced refractory solid tumors were treated with LY3321367 (mAb anti-TIM-3) alone or in combination with LY3300054 (mAb anti-PD-L1) and first results showed that both therapies were well tolerated [119]. LY3321367 treatment-related adverse events observed in two or more patients included pruritus, rash, fatigue, anorexia, and infusion-related reactions.

Preliminary signs of antitumor activity and good safety have been also reported in a phase I/II study on patients with advanced solid tumors with the combination of sabatolimab (mAb binding TIM-3) and spartalizumab (anti-PD-1) [120].

### 3.4. V-Domain Ig Suppressor of T Cell Activation

VISTA was the first described member of immunoglobulin superfamily ligand able to downregulate T cell responses in mice and in humans [44,121]. It is a type I transmembrane protein consisting of a single N-terminal IgV-domain, a transmembrane domain, and a cytoplasmic tail. The structural analysis, shows a similarity to the extracellular domain of the PD-L1 and PD-L2 and an analogous homology sequence between the IgV-domain and the B7 families [122]. However, VISTA does not express ITIM or immunoreceptor tyrosine-based switch motif (ITSM) conversely to B7 proteins family [123].

The role of VISTA in immune regulation is complicated and debated; it seems to act both as a ligand expressed on APC, and as a receptor on T cells. Evidence supports VISTA both as an immune checkpoint receptor suppressing T cell activation, proliferation, and cytokine production [123,124] and as a stimulatory checkpoint-like protein inducing anti-cancer immune responses [44,125,126,127]. Several studies have been made to identify the predictive and prognostic role of VISTA, but its role in some cases remains undefined [126,128,129,130,131]. The non-overlapping mechanisms of PD-L1 and VISTA make this combination an ideal strategy to overcome immune suppression, as shown in preclinical models where dual blockade had synergistic activity against T-cells favoring anti-tumor responses [124,132]. Based on these results, several clinical trials have been conducted.

A phase I clinical study are evaluating pharmacokinetics and safety of JNJ-61610588 (CI-8993), an anti-VISTA mAb, in patients with advanced cancers (NCT02671955). By inhibiting VISTA signaling, CI-8993 enhances T cell-mediated immune responses againts tumor cell growth. No study results have yet been reported.

CA-170 is an oral inhibitor that targets both VISTA and PD-L1 and showed remarkable antitumor effects and good safety in preclinical models. Results of a phase I clinical trial in patients with advanced solid tumor confirmed a favorable safety profile of CA-170 with preliminary evidence of anti-tumor activity [133]. Early results of a phase II study confirmed an excellent clinical benefit rate and PFS of CA-170 in different tumors, with a better safety benefit, compared to immune oncology antibodies. This is the first phase II study in which an oral immune agent exhibited activity in cancer patients, thus laying groundwork for evaluation in adjuvant and/or maintenance settings in non-squamous NSCLC [134].

### 3.5. Inducible T-Cell COStimulator (ICOS)

ICOS or CD278 is a CD28-superfamily costimulatory molecule involved in regulating T cell activation and adaptive immune responses [135,136,137]. ICOS and ICOS ligand (L) play an important role in memory-T and effector-T (Teff) cells development, although their role in cancer is still under investigation. The ICOS pathway has been shown to potentiate immunosuppression mediated by Tregs, although an antitumor effect related to the same pathway has been recognized [136]. Hence, new mAbs with agonistic or antagonistic function have been investigated in this setting [138,139].

Preclinical studies recorded a potentiated effect of anti-CTLA-4 with ICOS agonistic mAbs, showing antitumor superiority of the concomitant stimulation compared with anti-CTLA-4 alone [140]. Indeed, the use of agonistic or antagonistic Abs against ICOS alone appears to be less potent than anti CTLA-4 or PD-1/PD-L1 mAbs [141], whereas their combination is able to generate potent synergistic effects inhibiting the suppressive activity of T reg and potentiating the antitumour activity of T eff, including CD4+ and CD8+ subpopulations [140,142].

In the first clinical trial, INDUCE-1, feladilimab (ICOS-agonist Ab) alone or combined with pembrolizumab, showed a good tolerability and clinical activity profile on patients with advanced solid tumors [141]. In his study, the most frequent treatment-related AEs were fatigue (15%), fever (8%), elevation of hepatic enzymes (5%), also representing the most frequent grade 3–4 AE, and diarrhea (3%). One dose limiting grade 3 pneumonitis occurred.

Another ICOS agonist, vopratelimab (JTX-2011) showed in a phase I/II trial, to be well tolerated and to have antitumor effect in heavily pre-treated patients both alone and in combination with nivolumab (NCT02904226). Results of completed phase II trial testing vopratelimab plus CTLA-4 inhibitor in PD-1/PD-L1 inhibitor evaluated on patients with NSCLC or urothelial cancer are expected (NCT03989362).

Compared to ICOS agonist, ICOS antagonistic Abs have shown limited antitumour activity. Nevertheless, in a phase I/II study, KY1044 (anti-ICOS Ab), ICOS antagonists showed good tolerability both as single agents and combined with atezolizumab in solid tumors, including NSCLC [143].

Overall, data show thatf ICOS mAbs could play a critical role in effective responses to other ICIs, and that peripheral blood CD4+ ICOS^hi^ T cell subpopulations seem to be a promising biomarker of immune response [144].

### 3.6. B7 Homolog 3 Protein, (B7-H3)

B7-H3, also called CD276, is a member of the B7 family, comprising a short intracellular, a transmembrane and an extracellular domain [145,146]. Soluble B7-H3 isoform either is produced by alternative intronic splicing or is released from the cell surface through matrix metallopeptidase activity [147,148]. Initially B7-H3 was described as a positive co-stimulator that activates T cells and interferon-γ production, but recently has been reported as a factor involved in the inhibition of CD4+ and CD8+ T cells [149,150,151]. TREM-like transcript 2 (TLT-2) is considered as a potential receptor of B7-H3; however, the ambiguous roles of B7-H3 in the immune activity require further investigation to identify accurately its receptors [31,152].

The abnormal expression of B7-H3 could be considered an immune biomarker, and therefore a promising therapeutic immune checkpoint target. In clinical studies on cancer patients, high levels of B7-H3 expression were correlated with disease progression [153,154,155,156] specifically in NSCLC where B7-H3 expression is correlated with poor prognosis [153]. Recently, several mechanisms for targeting B7-H3 have been developed, although there are still doubts about the immunotherapeutic activity of B7-H3.

Enoblituzumab, an agent targeting B7-H3 was first evaluated in a phase I dose escalation study in patients with solid B7-H3 positive tumors. The treatment presented a favorable safety profile and an anti-tumor activity, since tumour shrinkage was obtained in several tumor types [157]. Results are expected from two completed phase I trials, testing the safety of enoblituzumab in combination with pembrolizumab (NCT02475213) or ipilimumab (NCT02381314) in patients with B7-H3 positive solid tumors.

Moreover, other approaches such as B7-H3 targeting antibody conjugated with deruxtecan or B7-H3 and CD3 dual-affinity retargeting protein are under evaluation in early phase study (NCT04145622).

### 3.7. B- and T-Lymphocyte Attenuator (BTLA)

BTLA or CD272 is a member of the CD28 coreceptor family [158] structurally and functionally similar to PD-1 and CTLA-4 [159]. It is formed by a single extracellular domain, a transmembrane region and a cytoplasmic domain that mediates a negative signaling to T cells by recruiting the small heterodimer partner 1 and 2 [160,161]. The binding between BTLA and its ligand Herpes virus entry mediator (HVEM), triggers the inhibition of T cell proliferation and cytokine production. CD160 competes with BTLA for the same binding site of HVEM, cysteine-rich domains (CRD) 1 and CRD2 and negatively regulates T cells, whereas LIGHT independently binds the opposite side of HVEM (CRD2/CRD3 region) [162,163,164] and represents a costimulatory molecule [163,165].

Song et al. confirmed that BTLA plays an essential role in immune cell infiltration and could function as a prognostic biomarker [166]. As member of the CD28 receptor family, BTLA can inhibit T cells when bound to HVEM. OX40 receptor, a member of the tumor necrosis factor receptor family like HVEM, is under investigation to prevent its interaction with BTLA.

Several agents targeting OX40 are under investigation in clinical trials. NCT04198766 is a phase I study designed to determine the safety profile and the maximum tolerated dose of INBRX-106 (OX40 agonist Ab), as a single agent or in combination with pembrolizumab in patients with solid tumors. Results of a completed phase I study evaluating the tolerability of PF-04518600 (OX40 agonist Ab) alone or in combination with PF-05082566 (4-1BB agonist) (NCT02315066) are expected, whereas cudarolimab (anti-OX40 Ab) alone or combination with sintilimab (anti-PD-1) is being tested in patients with advanced solid tumors (NCT03758001). The general characteristics of emerging immune checkpoint molecules are summarized in Table 1.

### 3.8. Other Immuno-Target Molecules

Other molecules have been identified as possible immune response regulators. Among these, Indoleamine 2,3-dioxygenase 1 (IDO1) is an immunomodulatory enzyme produced by alternately activated macrophages and other immunoregulatory cells [168].

IDO1, expressed in all cell of TME, has a crucial role in the aminoacid tryptophan to kynurenine degradation [169]. Its cytosolic expression is induced by IFN-γ, TNF-α, TGF-β, and other pro-inflammatory signals. By reducing tryptophan and favoring the increase in its metabolites, IDO1 suppresses T eff and NK cells and generates Treg and myeloid-derived suppressor cells [134,170]. Moreover, IDO1 supports tumor angiogenesis, antagonizing the anti-angiogenic effect of IFN-γ [171] and favoring tumor immune escape [172].

Epacadostat, a selective oral IDO1 inhibitor, was shown to increase Teff and NK cells proliferation and reduce Treg activation, particularly when it is associated with others immune checkpoint inhibitors [173]. Clinical studies confirmed the tolerability and anti-tumor activity of epacadostat in various advanced solid tumors [174,175]. In a phase I/II trial (ECHO-202/KEYNOTE-037), epacadostat plus pembrolizumab showed good tolerability and antitumor activity in different solid tumors, including pretreated advanced NSCLC [176]. Although the combination of epacadostat with pembrolizumab failed its primary endpoint in the phase III ECHO 301/KEYNOTE 252 in advanced melanoma [177], two completed phase III trials evaluated epacadostat plus pembrolizumab ± platinum-based chemotherapy in metastatic NSCLC at the forefront (NCT03322540, NCT03322566).

Navoximod, another investigational IDO1 inhibitor, showed acceptable tolerability and pharmacokinetics when combined to atezolizumab in advanced solid tumors including NSCLC, though lacking a clear evidence benefit [178]. Finally, a phase I study tested indoximod, a tryptophan-mimicking agent blocking mTORC1 (a protein with immunosuppressive role on T-cells) [179], combined with docetaxel and Tergenpumatucel-L, in advanced NSCLC. This trial is terminated for paucity of enrollment and due to the changing of standard of care (NCT02460367).

CD94/NK group 2 member A (NKG2A) is a cell surface glycoprotein that form disulfide-bonded heterodimers with CD94 and bind the MHC class Ib molecule HLA-E. It is usually expressed by NK cells, but also on T cells particularly on CD8+ [180]. HLA-E expression explains immunosuppressive action when bound by NKG2A [181] and its overexpression on tumor cells has been associated with poor outcomes [182].

Monalizumab, a mAb anti NKG2A, showed encouraging antitumor activity in early clinical studies. In a phase II trial, monalizumab in combination with durvalumab versus anti-PD-L1 alone showed improvements in ORR and PFS in stage III unresectable NSCLC in patients who did not progress to concomitant chemo- or radiotherapy [183].

CD73, known as ecto-5′-nucleotidase, is a novel immune checkpoint associated with adenosine metabolism that indirectly promotes tumor progression by suppressing antitumor immune response and promoting angiogenesis [184] In several tumors, including lung cancer, CD73 is upregulated, and its higher expression is associated with poor outcomes [184,185,186,187]. Preclinical evidence reported synergistic anti-tumor effects with anti-CD73 and PD-1/PD-L1 antibodies, due to a greater increase in intra-tumoral infiltration of CD8+ T cells [188,189].

Oleclumab, a mAb against CD73, was tested in association with durvalumab, versus durvalumab alone, after chemoradiation in unresectable stage III NSCLC. Oleclumab showed to improve PFS and to have a manageable safety profile [190]. In a phase II study, NeoCOAST durvalumab alone or combined with oleclumab, monalizumab or danvatirsen (the anti-STAT3 antisense oligonucleotide) has been evaluated as neoadjuvant therapy in patients with previously untreated, resectable, stage I-IIIA NSCLC [191]. One cycle of durvalumab in combination with other agents improved pCR and major pathological response rates versus durvalumab alone whit the same safety proflile. A correlation between response rate and baseline tumor CD73 and PD-L1 expression levels has been documented. Promising results, led to a phase II NeoCOAST-2 trial that are enrolling patients with resectable, early-stage NSCLC, to receive neoadjuvant durvalumab plus chemotherapy plus oleclumab followed by adjuvant durvalumab plus oleclumab, or neoadjuvant durvalumab plus chemotherapy and monalizumab, followed by adjuvant durvalumab plus monalizumab (NCT05061550).

Several immune checkpoint targets and their inhibitory molecules, alone or in combination with PD-1/PD-L1 and CTLA-4 inhibitors are under investigation in a multitude of ongoing clinical trials (Table 2).

## 4. Future Perspective

The revolutionary achievement of immunotherapy in treating lung cancer still presents several major challenges. First, only about 30% of patients with metastatic NSCLC and 20% of patients with extensive stage SCLC derive lasting benefits from ICIs. Presumed mechanisms of primary and acquired resistance to ICIs involve cancer intrinsic mechanism (i.e., defects in antigen presentation, altered ability to respond to interferon gamma signaling; oncogenic pathways) or cancer-extrinsic mechanism (i.e. exhausted or dysfunctional T cells; immunosuppressive pathways, altered metabolism and increased adenosine production) [192]. Second, the search for fully reliable biomarkers that predict the ICI response is still unsatisfactory, and PD-L1 expression, together with microsatellite instability, remain the only biomarkers used in clinical practice to date. The intertumoral and intratumoral heterogeneity of PD-L1 expression, the different tests to measure PD-L1 expression and the cutoff values used to define its positivity, make this biomarker imperfect [44,193,194,195,196]. Tumour mutation burden (TMB) has been proposed as a biomarker able to differentiate ICIs responders from non-responders in lung cancer, but wasn’t considered reliable enough to be routinely used in clinical practice [197].

Clinical needs move our research to improve survival, maximizing the results and limiting the rate of failure. New immune checkpoints inhibitors, stimulatory molecules, and combination treatments able to reduce tumor burden and improve durable antitumor response and survival have been evaluated. Several new ICIs such as anti LAG3, anti TIGIT, and anti TIM3 had the most promising results, especially when combined with anti-PD1/PD-L1 and CTLA-4.

Moreover, the inhibition of immunosuppressive agents such us NKG2A, CD73 and IDO1 showed synergic activity with different ICIs. Their actual role remains to be defined, as well as the benefit of adding them to ICIs. In addition to the identification of new molecules, examination of potential predictive biomarkers deriving from tissue, blood, microbiota, and tumor is crucial to identify responders and customize therapy, avoiding useless and harmful treatments [198].

New combinations aim for complementary approaches to restore tumoricidal activity of T cells, instead of relying solely on the well-established mechanism of checkpoint inhibition. Bispecific antibodies, oncolytic viruses, adoptive cell transfer therapy (ACT), vaccines and cytokines are under investigation combined with checkpoint inhibitors [199,200,201].

Specifically, in lung cancer, recent evidence suggests that macrophages are linked to immunosuppression, angiogenesis, and inflammation process. Preclinical and clinical studies are ongoing to evaluate the activity of inhibitory drugs, which limit macrophage recruitment and restore the antitumor phenotype [202].

Hence, a deep understanding of the tumor microenvironment is the key to improve lung cancer management not only through the implementation and optimization of immunotherapy, but also by strengthening the role of available options such as local treatments.

## 5. Conclusions

Immune checkpoint blockades lead to impressive and durable responses in cancer therapy, mainly for the treatment of unresectable, metastatic, and recurrent disease. At present, both PD-1/PD-L1 and CTLA-4 blockades have been approved for the treatment of lung cancer; however, the limited efficacy and immune-related adverse events of ICIs, led to the discovery of novel checkpoints molecules. These, including LAG-3, BTLA, TIM3, B7-H3, ICOS, VISTA, TIGIT can inhibit T-cell responses and are being investigated as therapeutic targets for immune checkpoint blockade. These promising targets, especially when combined with antibodies against PD-1/PD-L1, could overcome the limitations associated with the use of the currently approved ICIs. Several trials are underway, in order to obtain more robust data to incorporate these agents into clinical practice.

## Figures and Tables

**Figure 1 cancers-14-06145-f001:**
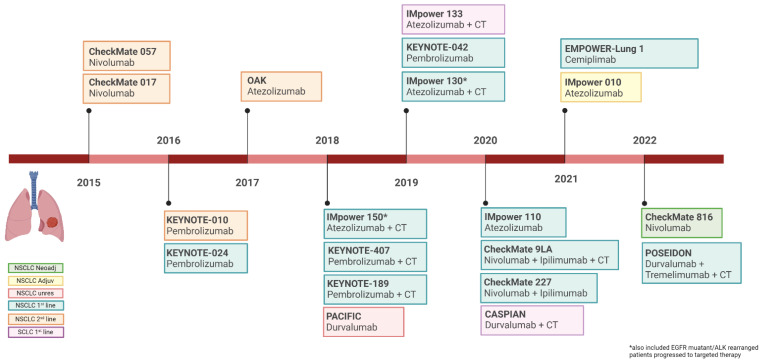
Timeline of Food and Drug Administration approval of immune checkpoints inhibitors in lung cancer. Image created with BioRender.com, accessed on 5 July 2022.

**Figure 2 cancers-14-06145-f002:**
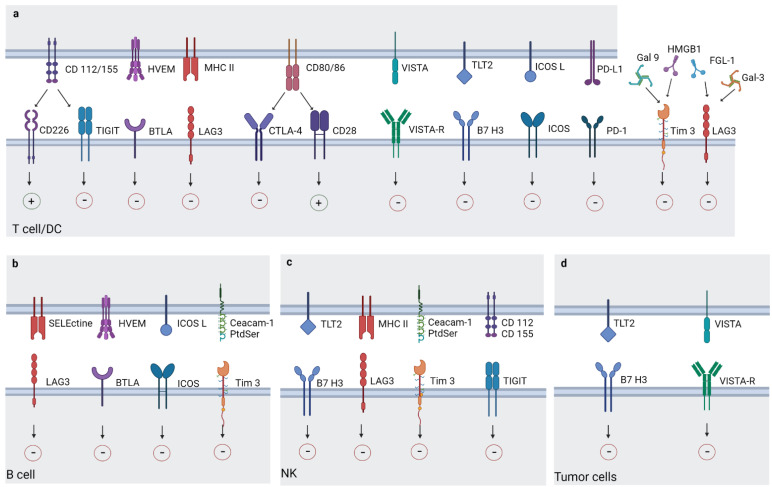
Expression of immune inhibitory checkpoints on different immune cells and tumor cells. Various immune checkpoints expressed by T cells and DC (**a**), B cells (**b**), B cells and NK (**c**), NK (**d**) and tumor cells. The figure shows the ligand for inhibitory receptors discussed in the review, the eventually co-stimulatory receptors activated by the ligands, and the contradictory or undefined roles of some of these molecules. Dendritic cells, (DC); natural killer, (NK); lymphocyte activation gene-3, (LAG-3); T cell immunoglobulin and mucin-domain containing-3, (TIM-3); T cell immunoreceptor with Ig and ITIM domains, (TIGIT); V-domain Ig suppressor of T cell activation, (VISTA); B7 homolog 3 protein, (B7H3); inducible T cell costimulatory, (ICOS; B and T cell lymphocyte attenuator (BTLA). Image created with BioRender.com. and accessed on 16 May 2022.

**Table 1 cancers-14-06145-t001:** Emerging immune checkpoint molecules and their ligands.

Protein	Gene Location (Human)	Expression Cell	Ligand and Presenting Cell		Immune Effect	Ref.
**LAG-3**	Chromosome 12p13.32	T cells NK B cells DCs	MHC II	APC	Reduction in T helper 1 (Th1) cell activation, proliferation, and cytokine secretion	[70,71,72,73,74,75,76,77,78,79]
			Galactine-3	Soluble		
			LSECtin	Tumor cells		
			FGL-1	Soluble		
**TIGIT**	Chromosome 3q13.31	T cells NK	CD 155 CD 112 Nec3	APC	Suppression of T cell and NK activation	[93,94,95,96,97,98]
			Fap2	*Fusobacterium nucleatum*		
**TIM-3**	Chromosome 5q33.2	T cells B cells DCs NK Monocyte Macrophages	Galectine-9	Soluble	Negative regulation of T cell responses	
			HMGB1	Soluble		[28,110,111,112,113,114,115,116,117]
			Ceacam	Unknown		
			PtdSer	Unknown		
**VISTA**	Chromosome 10q22.1	DCs Macrophage Monocytes T cells Tumor cells	VISTA-L	APC	Suppression of T cell activation, proliferation, and cytokine production Activation of anti-cancer immune responses.	[121,122,123,124,125,126,127]
**ICOS**	Chromosome 2q33.2	Activated memory T cells	ICOS-L	APC Somatic cells	Suppression of antitumor T cell response	[135,136,137,138,167]
**B7-H3**	Chromosome 15q24.1	APC NK T cells Monocytes Tumor cells	TLT2 (receptor)	Unknown	Co-stimulation of T cells activations Inhibition of CD4+ and CD8+ T cells	[145,146,147,148,149,150,151,152]
**BTLA**	Chromosome 3q13.2	B cells T cells DCs Macrophages	HVEM	Unknown	Inhibition of T cell proliferation and cytokine production	[158,159,160,161,162,163,164,165]

**Table 2 cancers-14-06145-t002:** Ongoing trial with new immune checkpoints targets in lung cancer.

Immune Check Point	No. of Trial	Status	Phase	Estimated Enrollment	Tumor Types	Setting	Investigated Agents	Primary End Points
**LAG-3**								
	NCT03625323	Active not recruiting	II	183	NSCLC, HNSCC	Untreated, unresectable or metastatic	Eftilagimod Alpha (anti-LAG3 mAb) + Pembrolizumab	ORR
	NCT04618393	Recruiting	I/II	43	Solid Tumors	Advanced	EMB-02 (anti-PD-1/LAG-3 bispecific mAb)	AEs and, AEs, ORR
	NCT03459222	Recruiting	I/II	184	Malignant Tumors	Advanced	Relatlimab (anti-LAG-3 mAb) + Nivolumab and BMS-986205 (IDO1 inhibitor) or Ipilimumab	AEs and AEs, DLT, ORR, DCR, mDOR
	NCT04140500	Recruiting	I	320	Solid Tumors, Melanoma, NSCLC, ESCC	Advanced and/or metastatic	RO7247669 (anti-PD-1/LAG3 bispecific Ab)	DLTs, ORR, DCR, DOR, PFS
	NCT04374877	Recruiting	I/Ib	220	RCC, HCC, NSCLC	Advanced	SRF388 (anti-IL-27 mAb)	DLT, ORR, AEs, ORR
	NCT03625323	Active, not ecruiting	II	189	NSCLC, HNSCC	Untreated unresectable or metastatic	Eftilagimod Alpha (soluble LAG-3 fusion protein) + Pembrolizumab	ORR
	NCT03250832	Active, not recruiting	I	111	Solid Tumors	Advanced	TSR-033 (anti-LAG-3 mAb) ± anti-PD-1	Safety, ORR
**TIGIT**								
	NCT04995523	Recruiting	II	147	NSCLC	Advanced or metastatic	AZD2936 (anti-TIGIT/PD-1 bispecific Ab)	AEs, ORR
	NCT04952597	Recruiting	II	120	SCLC	Untreated limited stage	Ociperlimab (anti-TIGIT) + Tislelizumab + CT	PFS
	NCT04746924	Recruiting	III	605	NSCLC	Untreated locally advanced, unresectable, or metastatic	Ociperlimab (anti-TIGIT) + Tislelizumab	PFS, OS
	NCT04294810	Recruiting	III	560	NSCLC	Untreated locally advanced, unresectable, or metastatic	Tiragolumab (anti-TIGIT) + Atezolizumab	PFS, OS
	NCT04791839	Recruiting	II	30	NSCLC	Untreated advanced	Zimberelimab (anti-PD-1) + Domvanalimab (anti-TIGIT) + Etrumadenant (anti-A2R)	ORR,
	NCT04262856	Recruiting	II	150	NSCLC	Metastatic	Zimberelimab (anti-PD-1) ± Domvanalimab (anti-TIGIT) ± Etrumadenant (anti-A2R)	ORR, PFS
	NCT04761198	Recruiting	I/II	125	Solid tumors	Locally advanced or metastatic	Etigilimab (anti-TIGIT) + Nivolumab	ORR
	NCT04736173	Recruiting	III	625	NSCLC	Locally advanced or metastatic	Zimberelimab (anti-PD-1) ± Domvanalimab (anti-TIGIT)	OS, PFS
	NCT03739710	Recruiting	II	140	NSCLC	Relapsed/refractory advanced	Feladilimab, Ipilimumab (anti-CTLA-4), GSK4428859A, Dostarlimab (anti-PD-1) (various combination versus SoC)	AEs, DLT, OS
	NCT04995523	Recruiting	I/II	147	NSCLC	Advanced, or metastatic	AZD2936 (anti-TIGIT/anti-PD-1 bispecific Ab)	AEs, ORR
	NCT04746924	recr	3	605	NSCLC	Untreated PD-L1-selected, and locally advanced, unresectable, or metastatic	BGB-A1217 (anti-TIGIT Ab) + Tislelizumab	PFS, OS
	NCT04585815	Recruiting	I and II	375	NSCLC	Advanced or metastatic	Sasanlimab (anti-PD-1) + Encorafenib and Binimetinib or Axitinib and SEA-TGT (anti-TIGIT)	DLT, ORR
	NCT04866017	Recruiting	III	900	NSCLC	Locally advanced, unresectable	Tislelizumab (anti-PD-1) ± Ociperlimab (anti-TIGIT) + CRT	PFS, CRR
	NCT04294810	Recruiting	III	635	NSCLC	Untreated locally advanced, unresectable, or metastatic PD-L1-selected	Tiragolumab + Atezolizumab	PFS, OS
	NCT05102214	Recruiting	I/II	150	Solid Tumors NSCLC	Locally Advanced or Metastatic	HLX301 (PDL1/TIGIT bispecific Ab)	
	NCT04791839	Recruiting	II	30	NSCLC	Previously treated	Zimberelimab + Domvanalimab (anti-TIGIT) and Etrumadenant	ORR, PR
	NCT05014815	Recruiting	2	270	NSCLC	Untreated locally advanced, unresectable, or metastatic	Ociperlimab (anti-TIGIT) and Tislelizumab + CT	PFS
	NCT05060432	Recruiting	I/II	376	Lung Cancer, Head and Neck cancer, Melanoma	Advanced	EOS-448 (anti-TIGIT) + SoC or Investigational Therapies	DLT, AE, ORR, RP2D
	NCT04952597	Active, not recruiting	II	126	SCLC	Limited Stage	Ociperlimab + Tislelizumab + CRT	PFS
	NCT03563716	Active, not recruiting	II	135	NSCLC	Chemotherapy-naïve patients with locally advanced or metastatic	Tiragolumab, (anti-TIGIT) + Atezolizumab	ORR, PFS
	NCT04256421	Active, not recruiting	III	490	SCLC	Untreated Extensive Stage	Atezolizumab + Carboplatin and Etoposide ± Tiragolumab (anti TIGIT)	PFS, OS
	NCT04672356	Active, not recruiting	1	20	Lung Cancer	Advanced	IBI939 (anti-TIGIT Ab) + Sintilimab	AE, RP2D
	NCT04672369	Active, not recruiting	1	42	Lung Cancer	Advanced	IBI939 (anti-TIGIT Ab) + Sintilimab	ORR
**TIM-3**								
	NCT04931654	Recruiting	II	81	NSCLC	Advanced or metastatic	AZD7789 (PD-1/TIM-3 bispecific Ab)	AE, DLT, ORR
	NCT03744468	Recruiting	II	162	HNSCC, NSCLC, RCC	Advanced	BGB-A425 (anti-TIM-3) and LBL-007 (anti-LAG-3) + tislelizumab	MTD, ORR
	NCT03708328	Active, not recruiting	I	134	Solid Tumors, Melanoma, NSCLC, SCLC, ESCC	Advanced and/or Metastatic	RO7121661 (anti-PD-1/TIM-3 bispecific Ab)	Dose Escalation, ORR, DCR, DOR, PFS
	NCT02817633	Recruiting	I	396	Solid Tumors	Advanced	TSR-022, (anti-TIM-3 Ab)	DLTs, AEs, ORR
**B7-H3**								
	NCT04432649	Recruiting	I/II	100	Solid Tumor	Refractory and/or recurrent	4SCAR-276 (anti-B7-H3)	AE
	NCT05280470	Recruiting	II	80	SCLC	Pretreated Extensive Stage	DS-7300a (anti-B7-H3 ADC)	ORR
	NCT03729596	Recruiting	I/II	182	Solid Tumor, SCCHN, TNBC, Melanoma, mCRPC, NSCLC	Advanced	MGC018 (anti-B7-H ADC) ± MGA012 (anti-PD-1)	AE and SAE, DLT
**VISTA**	NCT05082610	Recruiting	I	240	Solid Tumor, NSCLC, TNBC	Advanced	HMBD-002-V4C26 (anti-VISTA) ± Pembrolizumab	DLT, Safety
**BTLA**								
	NCT04137900	Recruiting	I	499	Solid Tumors	Advanced	TAB004 (anti-BTLA) ± Toripalimab	TRAE
	NCT03758001	Active, not recruiting	I	38	Solid Tumor	Advanced	Cudarolimab (anti-OX40) + Sintilimab (anti-PD-1)	AEs
	NCT05000684	Recruiting	I/II	66	Lung Cancer	Advanced	JS004 (anti-BTLA) injection + Toripalimab	AE
**ICOs**	NCT03829501	Recruiting	I/II	208	Solid Tumors	Advanced	KY1044 (anti-ICOS) ± Atezolizumab	AEs, ORR, DLTs

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
