# Peer review of "Lung Cancer Immunotherapy: Beyond Common Immune Checkpoints Inhibitors"

_cancers, 2022, doi:10.3390/cancers14246145_

Round 1

Reviewer 1 Report

This paper reviewed the efficacy and limitation of immunotherapy in lung cancer, especially these with the innovative immune targets. Furthermore, the perspective of the innovative immunotherapies are discussed. The topic is interesting and fits the scope of Cancers. It may also benefit the development of novel immunotherapy for patients with lung cancer. In general, the manuscript is well-organized, and the references are supportive to the conclusions. The key issues are required to be solved before its publication on Cancers.

1. The (potential) immune-related adverse events (irAEs) of the incoming new immunotherapy are suggested to be discussed (separate discussion with the individual new therapy, or summarized discussion in a separate section).

2. The repeating description of the contents in Tables is required to be shortened again, especially in section 3 for Table 3.

Author Response

This paper reviewed the efficacy and limitation of immunotherapy in lung cancer, especially these with the innovative immune targets. Furthermore, the perspective of the innovative immunotherapies are discussed. The topic is interesting and fits the scope of Cancers. It may also benefit the development of novel immunotherapy for patients with lung cancer. In general, the manuscript is well-organized, and the references are supportive to the conclusions.

The key issues are required to be solved before its publication on Cancers.

Thank you for the positive comments and suggestions.

1.The (potential) immune-related adverse events (irAEs) of the incoming new immunotherapy are suggested to be discussed (separate discussion with the individual new therapy, or summarized discussion in a separate section).

Thanks for the suggestion with which we agree. Since the adverse events of the new immune checkpoint inhibitors are like the known ones, we preferred to add details in the individual sections rather than separately.

  1. The repeating description of the contents in Tables is required to be shortened again, especially in section 3 for Table 3.

Fully agreeing with what has been suggested, we have tried to avoid repeating the information in the table, in the text.

Reviewer 2 Report

1. In lines 26 - 29, suggest emphasizing that significant success has been achieved with anti-PD1/PDL1 monoclonal antibodies and that the combination of anti-CTLA 4 mabs + anti-PD1/PDL1 mabs are the only immunotherapy combination regimens shown to be effective and to be approved by regulatory agencies. 

2. In figure 1, suggest adding POSEIDEN  trial(Johnson M, J Clin Oncol 2022) which has led to FDA approval of chemotherapy combined with durvalumab and tremelimumab for first line treatment for stage IV NSCLC>.

3. Also in figure 1, Checkmate 227 compared ipi/nivo to chemotherapy alone, not + chemotherapy.

4. After line 172, suggest discussing results of POSEIDEN trial.

5. Also, it might be worth noting that 3 stage IV NSCLC trials which evaluated first line combination of anti-PD1/PDL1 and an anti-CTLA4 monoclonal antibodies were associated with superior overall survival compared to chemotherapy alone in patients whose tumors' PDL1 expression was < 1%. 

6. After line 188, it would be good to include brief section regarding the frequency of PDL1 expression: < 1% versus >1 to < 50% versus > 50% and regarding outcomes related to PDL1 expression is generally associated with superior outcomes with immunotherapy. 

7. If you include a biomarker section, it would be worth noting that 3 first line stage IV NSCLC trials have shown superior survival with combined anti-PD1/PDL1 and anti-CTLA4 regimens versus chemotherapy alone in patients whose tumors' PDL1 expression was > 1%. 

8. You have described the prognostic significance of some of the new immune targets. If data are available for the frequency of the new targets in NSCLC, it would be good to include this information in a biomarker section. 

9. Are biomarker determinations required in the trials listed in your review? 

Author Response

  1. In lines 26 - 29, suggest emphasizing that significant success has been achieved with anti-PD1/PDL1 monoclonal antibodies and that the combination of anti-CTLA 4 mabs + anti-PD1/PDL1 mabs are the only immunotherapy combination regimens shown to be effective and to be approved by regulatory agencies. 

Thanks for the suggestion, we've edited accordingly.

  1. In figure 1, suggest adding POSEIDEN  trial(Johnson M, J Clin Oncol 2022) which has led to FDA approval of chemotherapy combined with durvalumab and tremelimumab for first line treatment for stage IV NSCLC>.

Thank you,we have added the recently approved study in Figure 1.

  1. Also in figure 1, Checkmate 227 compared ipi/nivo to chemotherapy alone, not + chemotherapy.

Thanks, the correction has been made.

  1. After line 172, suggest discussing results of POSEIDEN trial.

In accordance with your suggestion, we have added data from the recently approved POSEIDON study.

Also, it might be worth noting that 3 stage IV NSCLC trials which evaluated first line combination of anti-PD1/PDL1 and an anti-CTLA4 monoclonal antibodies were associated with superior overall survival compared to chemotherapy alone in patients whose tumors' PDL1 expression was < 1%. (?)

Thanks for the comment. As suggested, in the new section... we discussed the results of monotherapy and combination according to PD-L1 expression.

  1. After line 188, it would be good to include brief section regarding the frequency of PDL1 expression: < 1% versus >1 to < 50% versus > 50% and regarding outcomes related to PDL1 expression is generally associated with superior outcomes with immunotherapy. 

As suggested, we have added a new section regarding the frequency of PD-L1 in lung cancer and the impact that PD-L1 expression has on the efficacy of immunotherapy.

You have described the prognostic significance of some of the new immune targets. If data are available for the frequency of the new targets in NSCLC, it would be good to include this information in a biomarker section. 

This comment is really interesting, however the data present in the literature, relating to the expression of the new targets are limited and still poorly defined.

  1. Are biomarker determinations required in the trials listed in your review? 

In the studies we mentioned, specific biomarkers for the new immune targets were not required

Reviewer 3 Report

Brief summary

This review introduces development of immunotherapy in lung cancer treatment research, as well as s summarizes the current role of immune checkpoints inhibitors in lung cancer immunotherapy and innovative immune checkpoints molecules which could be potential immunotherapy targets. Authors described comprehensively various immune checkpoints inhibitors for emerging immune checkpoint molecules and novel immune checkpoint molecules. The applications and limitations of current immunotherapy in lung cancer treatment are well enumerated and briefly discussed. Also, authors deeply illustrated mechanism of novel immune checkpoint molecules and presented clearly recent researches and clinical trials for novel immune checkpoint molecules in lung cancer. In addition, this review cites most relevant studies, most of which are recent publications. However, this review still omits some relevant references, thus more references need to be cited properly.

Altogether, I recommend publication after minor revisions as detailed in comments for authors.

Major issues

1)     This article shares some degree of similarity with article published recently (PMID: 33741032 ) in section of novel immune checkpoints molecules, thus PMID: 33741032 need to be cited or discussed properly in the manuscript.

2)     Introduction Page2 Line 57 “specifically non-small cell lung cancer (NSCLC)” : Ref 10 only mentioned lung cancer but not NSCLC, properly cite statement in ref 10.

3)     Introduction Page2 Line 57 “specifically non-small cell lung cancer (NSCLC)” : Ref 11 is same as the Ref 5. Please provide the correct references for this sentence.

4)     Introduction Page 2 Line 59 “a cornerstone of the treatment (2).” : what is (2)? Does (2) mean Ref 2?  

5)     Table 1 and Table 2: Shows the citations in Table 1 and Table 2 for letting reader know where this information are collected.

6)     Section 3.3 Page 11 line 357: Ref 100 is about function of VISTA ON anti-tumor immunity but not the TIM-3. Cites relevant references to support this sentence.

7)     Section 3.3 Page 11 Line 374-375 “Several studies have been made to……” : Ref 108 is not relevant to the VISTA. Please provide correct reference.

Minor issues

1)     Abstract Page1 Line 33: Add the full name of abbreviation ”TIGIT” for reader who might not be familiar with this abbreviation, as it is first time it appears in the text.

2)     Introduction Page 1 Line 45 “Tumor cells develop numerous……”: Add more relevant references, such as PMID: 17571260, PMID: 32610070 and, Parayath, Neha, et al.  Regenerative Engineering and Translational Medicine (2020)

3)     Introduction Page 2 Line 49 “However, the most affecting mechanism to……” : Add the primary references, since ref4 and ref5 are secondary references.

4)     Introduction Page 2 Line 72 “acquired resistance to ICIs”: add more relevant references about primary or acquired resistance to ICIs, such as PMID: 27951441 PMID: 25891173, PMID: 12407406, PMID: 2795144.

5)     Section 3.2 Page 10 line 316-319: Provide relevant references to support this paragraph, or clarify the information source.

6)     Section 3.4 Page 11 line 359 “VISTA was the first described……”: include the ref 100 for this sentence.

7)     Section 3.4 Page 11 line 373-374 “as stimulatory checkpoint-like protein……” : Cites relevant references such as PMID: 24691994, PMID: 29375120.

8)     Section 3.8 Page 15 line 539: it seems that this paragraph is more relevant to Table 2 rather than Table 1.  

Author Response

This review introduces development of immunotherapy in lung cancer treatment research, as well as s summarizes the current role of immune checkpoints inhibitors in lung cancer immunotherapy and innovative immune checkpoints molecules which could be potential immunotherapy targets. Authors described comprehensively various immune checkpoints inhibitors for emerging immune checkpoint molecules and novel immune checkpoint molecules. The applications and limitations of current immunotherapy in lung cancer treatment are well enumerated and briefly discussed. Also, authors deeply illustrated mechanism of novel immune checkpoint molecules and presented clearly recent researches and clinical trials for novel immune checkpoint molecules in lung cancer. In addition, this review cites most relevant studies, most of which are recent publications.

However, this review still omits some relevant references, thus more references need to be cited properly.

Altogether, I recommend publication after minor revisions as detailed in comments for authors.

Thanks for the comments supporting our work. Below we have tried to resolve the required points.

Major issues

  • This article shares some degree of similarity with article published recently (PMID: 33741032 ) in section of novel immune checkpoints molecules, thus PMID: 33741032 need to be cited or discussed properly in the manuscript.

In agreement with your comment we quoted the article at the beginning of the discussion on the new immune checkpoints.

2)     Introduction Page2 Line 57 “specifically non-small cell lung cancer (NSCLC)” : Ref 10 only mentioned lung cancer but not NSCLC, properly cite statement in ref 10.

Thanks for the suggestion, we have reworded the sentence and replaced the reference.

  • Introduction Page2 Line 57 “specifically non-small cell lung cancer (NSCLC)” : Ref 11 is same as the Ref 5. Please provide the correct references for this sentence.

The reference 11 has been replaced with the new reference 10 accordingly.

4)     Introduction Page 2 Line 59 “a cornerstone of the treatment (2).” : what is (2)? Does (2) mean Ref 2?  

  Sorry for the mistake, the reference omitted now corresponds to the new refs 11 and 12.

5)     Table 1 and Table 2: Shows the citations in Table 1 and Table 2 for letting reader know where this information are collected.

        As suggested, we have added the references in table 1. Table 2 shows the reference NCT for each study.

6)     Section 3.3 Page 11 line 357: Ref 100 is about function of VISTA ON anti-tumor immunity but not the TIM-3. Cites relevant references to support this sentence.

Thanks for the comment, there was an error in the reference given. The sentence has been omitted in the text as it is reported in table 2.

7)     Section 3.3 Page 11 Line 374-375 “Several studies have been made to……” : Ref 108 is not relevant to the VISTA. Please provide correct reference.

Thanks for the suggestion, we deleted the reference accordingly.

Minor issues

  • Abstract Page1 Line 33: Add the full name of abbreviation ”TIGIT” for reader who might not be familiar with this abbreviation, as it is first time it appears in the text.

Thanks, the full name of abbreviation has been reported.

  • Introduction Page 1 Line 45 “Tumor cells develop numerous……”: Add more relevant references, such as PMID: 17571260, PMID: 32610070 and, Parayath, Neha, et al.  Regenerative Engineering and Translational Medicine (2020)
  •  

 Thanks for the suggestions, the references relating to the sentence have been added in the text.

  • Introduction Page 2 Line 49 “However, the most affecting mechanism to……” : Add the primary references, since ref4 and ref5 are secondary references.

We added a primary reference, accordingly.

  • Introduction Page 2 Line 72 “acquired resistance to ICIs”: add more relevant references about primary or acquired resistance to ICIs, such as PMID: 27951441 PMID: 25891173, PMID: 12407406, PMID: 2795144.

Thank you for these suggestions. The appropriate references have been added, accordingly.

5)     Section 3.2 Page 10 line 316-319: Provide relevant references to support this paragraph, or clarify the information source.

 Thanks for the comment, the references relating to the sentence mentioned are given in the text.

6)     Section 3.4 Page 11 line 359 “VISTA was the first described……”: include the ref 100 for this sentence.

Thanks, reference has been included.

7)     Section 3.4 Page 11 line 373-374 “as stimulatory checkpoint-like protein……” : Cites relevant references such as PMID: 24691994, PMID: 29375120.

Thanks for your tips. The references have been reported as suggested.

8)     Section 3.8 Page 15 line 539: it seems that this paragraph is more relevant to Table 2 rather than Table 1.  

Thanks for the comment, indeed there was an error reporting the table number.